# Development and Clinical Trials of Nucleic Acid Medicines for Pancreatic Cancer Treatment

**DOI:** 10.3390/ijms20174224

**Published:** 2019-08-29

**Authors:** Keiko Yamakawa, Yuko Nakano-Narusawa, Nozomi Hashimoto, Masanao Yokohira, Yoko Matsuda

**Affiliations:** Oncology Pathology, Department of Pathology and Host-Defense, Faculty of Medicine, Kagawa University, 1750-1 Ikenobe, Miki-cho, Kita-gun, Kagawa 761-0793, Japan

**Keywords:** nucleic acid medicine, pancreatic cancer, clinical trial, siRNA, antisense oligonucleotide

## Abstract

Approximately 30% of pancreatic cancer patients harbor targetable mutations. However, there has been no therapy targeting these molecules clinically. Nucleic acid medicines show high specificity and can target RNAs. Nucleic acid medicine is expected to be the next-generation treatment next to small molecules and antibodies. There are several kinds of nucleic acid drugs, including antisense oligonucleotides, small interfering RNAs, microRNAs, aptamers, decoys, and CpG oligodeoxynucleotides. In this review, we provide an update on current research of nucleic acid-based therapies. Despite the challenging obstacles, we hope that nucleic acid drugs will have a significant impact on the treatment of pancreatic cancer. The combination of genetic diagnosis using next generation sequencing and targeted therapy may provide effective precision medicine for pancreatic cancer patients.

## 1. Introduction

Despite advances in diagnostics and therapeutics, the prognosis of pancreatic cancer remains poor with an overall five-year survival rate of 6%, due in part to difficulties in treating carcinoma at an advanced stage. Mutations of *KRAS*, *CDKN2a*, *TP53*, and *SMAD4* are driver mutations in pancreatic cancer; however, a targeted approach for those molecules has not been successful yet. Precision medicine for individual patient has been greatly expected to improve pancreatic cancer patients’ outcomes. Recent advances of comprehensive gene analysis using next-generation sequencers can provide a wealth of information of genetic abnormalities of cancers [1,2]. There have been several candidates for treatment targets in pancreatic cancer. Approximately 30% of pancreatic cancer patients harbor druggable mutations; for example, *KRAS*, *BRCA1* and *2*, *PALB2*, *ATM*, *HER2*, *MET*, *MLH1*, *MSH2*, *MSH6*, *PMS2*, *PI3CA*, *PTEN*, *CDKN2A*, *BRAF*, and *FGFR1* [2]. However, there has been no clinical therapy targeting these molecules, because it is difficult to inhibit target RNA in humans. 

RNA interference (RNAi) is a biological process in which RNA molecules inhibit gene expression or translation by neutralizing targeted mRNA molecules. Nucleic acid medicine consists of natural or chemically modified nucleotides that can act directly without changes in gene expression [3]. These drugs show high specificity and can target mRNA and noncoding RNAs. Nucleic acid medicine is considered the next-generation treatment next to small molecules and antibodies. There are several aspects of nucleic acid therapy that are potentially advantageous over traditional drugs. These include the ability to generate specific inhibitors of targets that were previously inaccessible, with the only limit being the genetic information available. Inhibition of mRNA expression has the potential to produce faster and longer-lasting responses than protein inhibition by conventional targeted therapy. Moreover, the side-effects of nucleic acid medicine might be less than those of conventional therapy [4]. Lastly, oligonucleotides can be chemically synthesized and thus their development duration is relatively short compared to antibodies. 

There are several kinds of nucleic acid drugs, including antisense oligonucleotides (ASOs), small interfering RNAs (siRNAs), microRNAs (miRNAs), aptamers, decoys, and CpG oligodeoxynucleotides (CpG oligos) (Table 1). They can be classified as either extracellular or intracellular according to their site of function; ASOs, siRNAs, miRNAs, and decoys act in the nucleus or cytoplasm, while aptamers bind to extracellular proteins and CpG oligos act on Toll-like receptor 9 (TLR9) in the endosome. The drugs also have different targets; ASOs, miRNAs, and siRNAs target RNA, whilst aptamers, decoys, and CpG oligos target proteins. Nucleic acid drugs are suited for coextinction or therapeutic synergy, which may represent an important step to overcome compensatory effects typically observed in cancer cells following knockdown of a single target. In this review, we provide an update on the current research of nucleic acid-based therapies, focusing on ASO and siRNA for pancreatic cancer, and summarize the outcomes from published data.

## 2. Functions 

### 2.1. Antisense Oligonucleotides

ASOs are single strands of DNA or RNA that are complementary to a chosen sequence. In the case of antisense RNA, they prevent protein translation of certain messenger RNA strands by binding to them [5].

Antisense DNA can be used to target a specific, complementary (coding or noncoding RNA). If binding takes place, this DNA/RNA hybrid can be degraded by the enzyme RNase H. After crossing the cell membrane, ASOs target mRNA directly in the nucleus or cytosol, thus blocking and neutralizing the targeted miRNA, with the help of the enzyme RNase H1. Furthermore, ASOs have various functions, including the inhibition of translation, miRNA, and splicing. ASOs have been investigated for more than 20 years and their use is now a standard technique in developmental biology and they are used to study altered gene expression and gene function. Recently, several ASOs have been modified for an unnecessary drug delivery system (DDS). 

### 2.2. siRNAs

siRNAs are double-stranded RNAs with a length of 20–25 base pairs. siRNAs can suppress the gene expression via sequence specific inhibition of RNA expression (RNA interference, RNAi). The cellular process of RNAi occurs in almost all eukaryotic organisms [6]. After being processed by the ribonuclease III-like DICER enzyme, siRNA interacts with RNA-induced silencing complex to block and neutralize the target mRNA [7]. siRNA libraries have been created to dissect the function of independent genes since they show high sequence specificity. The application of siRNAs allows researchers to discover novel targets and pathway mediators. 

### 2.3. Aptamers

Nucleic acid aptamers are short single-stranded DNA or RNA oligonucleotides that fold into unique three-dimensional structures and bind to a wide range of targets, including proteins, small molecules, metal ions, viruses, bacteria, and whole cells [8]. Aptamers have high specificity and binding affinities (in the low nanomolar to picomolar range) similar to those of antibodies and are frequently referred to as ‘chemical antibodies’. Proteins constitute by far the largest class of aptamer targets. The high stability of aptamer–protein complexes, frequently characterized by a Kd in the low nanomolar range, combined with an excellent specificity of interaction make aptamers valuable tools for various applications, such as affinity purification, bio-sensing, imaging, and enzyme inhibition [9].

### 2.4. Decoys

Decoys are double-stranded molecules that mimic the consensus DNA binding site of a specific transcription factor in the promoter region of its target genes [10]. The regulation of transcription of disease-related genes in vivo has important therapeutic potential. Gene expression controlled by the transcription factor is effectively prevented, thereby effectively silencing gene expression and preventing protein production. Therefore, being less specific in comparison with the siRNA or ASO method, the decoy technique can be considered a gene silencing approach. 

### 2.5. CpG Oligos

CpG oligodeoxynucleotides (CpG oligos) are short single-stranded synthetic DNA molecules that contain cytosine triphosphate deoxynucleotide followed by a guanine triphosphate deoxynucleotide [11]. Synthetic phosphorothioate oligodeoxynucleotides bearing unmethylated CpG motifs can mimic the immune-stimulatory effects of bacterial DNA and are recognized by Toll-like receptor 9 (TLR9), which is constitutively expressed only in B cells and plasmacytoid dendritic cells. Nucleotide modifications at positions at or near the CpG dinucleotides can severely affect immune modulation. CpG oligos induce type I interferon, cytokines, B cell proliferation, dendritic cell maturation, and natural killer cell activation. CpG oligos have been applied for antiallergenic or anticancer treatment. 

## 3. Modifications of Nucleic Acid Drugs

Although, the function of nucleic acid drugs is promising, several challenges have been identified, including lack of stability against extracellular and intracellular degradation by nucleases, poor uptake and low potency at target sites of nucleic acid drugs, and off-target effects [12]. Off-target effects are nonspecific suppressive effects of nucleic acid drugs. Although it has been considered that nucleic acid drugs possess high specificity, several nucleic acid drugs can affect gene expression of multiple genes. Furthermore, nucleic acid drugs are quickly degraded by RNase in vivo. In humans, naked nucleic acid drugs preferentially accumulate in the liver and kidneys, which causes the nucleic acid drugs to be rapidly cleared from circulation with poor tissue distribution [13]. The pursuit of clinically viable antisense drugs has led to the development of various types of strategies, such as carriers or chemical modifications. Apart from structural modification of oligonucleotides, different cell-penetrating peptides and ligands conjugated to oligonucleotide-based DDS are normally adopted following the conjugation. 

### 3.1. Structural Modifications of Nucleic Acid Drugs

Important modifications have been implemented to improve the therapeutic potential of nucleic acid medicines. However, the properties of the modifications have also led to some decreased affinity for the target sequence, with associated nonhybridization toxicities such as complement activation, increased coagulation times, or immune activation (Table 2). Another concern relates to the hybridization-dependent toxicity, caused by exaggerated action of the drug or off-target hybridization. 

The first of the modifications included phosphorothioate backbone modification, which defined the first-generation nucleic acid drugs [5]. One of the nonbridge oxygen atoms in the diester bond is replaced by sulfur. Chemical modification can help enhance cellular uptake and increase the bioavailability of the modified nucleic acid drugs. Resistance to circumscribed nucleases is also effectively increased. However, although the modified siRNA is found to be significantly stable in the body, it increases the cytotoxicity and decreases the gene silencing effect. Modification of phosphorylated phosphate ester in the phosphorylation location damages RISC activity [14].

The second-generation nucleic acid drugs included the nucleoside analogues containing a modified sugar moiety, such as 2′-O-methyl-modified or 2′-O-methoxyethyl. The 2′ modifications inhibit the ability of RNase H to cleave the bound sense RNA strand within the heteroduplex formed between the nucleic acid drugs and the target RNA [15]. The widespread use of thiophosphate modifications results in a certain cytotoxicity, but the 2′-O-methylation improves the siRNA activity and is nontoxic to normal cells [16]. The activity of siRNA depends on the position of the modified parts.

Base modification plays an important role in the function of nucleic acid drugs; for example, it can improve the function of siRNA and increase the ability of the siRNA interaction with the target mRNA. The modification increases the ability of RISC to recognize and cleave the mRNA. The modifications on the base include adenine methylation and deamination, cystosine methylation, hydroxymethylation and carboxyl substitution, and guanine oxidation, etc. [17]. The modified bases are related to the changes of functional groups, which is the basis of triggering the functional changes through the modification of structure of nucleic acid drugs.

Oligonucleotide analogs’ replacement includes peptide substitution, and the resulting materials typically include peptide nucleic acid, locked nucleic acid, and morpholino phosphamide. They can reduce the degradation of oligonucleotides by nucleases, and have low toxicity and a slight decrease in affinity compared with unmodified sequences [18]. These nucleotide analogs do not support the cleavage of RNase H-mediated target mRNA in ASPs; thereby, they primarily exhibit their reflective activity by steric hindrance to prevent gene expression during transcription or translation. This method further enhances the binding affinity, nuclease resistance, and targeted effect compared with several other chemical modifications. 

### 3.2. Conjugation of Ligand or Cell-Penetrating Peptides

Cell-penetrating peptides are a class of short peptides that are rich in cations and can efficiently enter cells through penetrating biofilms. Based on these properties, cell-penetrating peptides are used to modified DNA, RNA, and oligonucleotides and are loaded on nanocarriers for therapy. The conjugation of oligonucleotides and cell-penetrating peptides can overcome the deficiencies of cytotoxicity and enhance the efficiency in eukaryotic cells. Complexes formed by cationic cell-penetrating peptides and anionic oligonucleotides which are formed through electrostatic interaction can promote oligonucleotides’ entry into cells and initiate RNA interfering, leading to silencing of endogenous genes [19]. Cell-penetrating peptides include cysteine, transactivator of transcription peptide [20], and gelatin [21]. 

## 4. Aptamers

Aptamers are synthetic single-stranded oligonucleotides of short length (20–100 nucleotides) whose three-dimensional disposition confers high avidity for their target DNA or RNA. They shows high stability, lack of immunogenicity, flexible structure, and small size, which increases their penetration strength [22]. Aptamer-based targeted delivery of siRNAs using aptamer–siRNA chimeras are becoming a very useful tool for targeting gene-knockdown in cancer therapy [23]. Aptamer–siRNA chimeras bind the aptamer’s receptor and upon engagement, the chimera–receptor complex is embedded into an endocytosis vesicle. The chimera reaches the cytoplasm and the duplex siRNA is recognized by Dicer and loaded into Dicer and RNA-induced silencing complex (RISC). Several aptamers have been reported for treatment of prostate, breast, and colon cancer, melanoma, lymphoma, and glioblastoma, for example *PSMA*, *4-1BBm EpCAP*, *CTLA4*, *PDGFRβ*, *HER2*, and *HER3* [23].

## 5. Drug Delivery Systems of Nucleic Acid Drugs

DDS has been necessary to regulate the drug distribution in the body in terms of quantity and spatiotemporal aspects. Several kinds of DDSs have been developed based on the diameter of medicine, specific antibody for tumor, sustained release, and percutaneous absorption. They are expected to improve the specificity, effects, usability, and economy of drug as well as to suppress the side-effects. 

Various carriers of siRNAs have become increasingly available because RNAi can integrate short hairpin RNA into the cell genome, leading to stable siRNA expression and long-term knockdown of a target gene. Nonviral carriers have been increasingly preferred owing to lower toxicity compared with other carrier methods. These carriers typically involve a positively charged vector (cationic cell-penetrating peptides, cationic polymers, and lipids), small molecules (cholesterol, bile acids, lipids, and PEGylated lipids), polymers, antibodies, aptamers, and lipid and polymer-based nanocarriers encapsulating the siRNA [24]. Specific delivery of siRNAs to hepatocytes has been accomplished by conjugation to *N*-acetylgalactosamine in order to target an asialoglycoprotein receptor present in the liver [25]. 

Different nanocarrier strategies are still needed in practical applications to make them more effective in diagnosing and treating diseases. A combination of chemical modification and a nanoparticle-based DDS is likely to be more effective for oligonucleotide delivery. For example, the siRNA can be modified with the free thiol group of the amino acid cysteine on cell-penetrating peptides, then they are encapsulated into ultrasound-sensitive nanomicrobubbles. When nanomicrobubbles reach the target site, they disintegrate under external ultrasonic irradiation, releasing siRNA to achieve cytoplasmic delivery [26]. 

Liposomes are widely used as oligonucleotide delivery systems (Table 3). Cationic liposomes include monovalent lipids such as DODMA and DOTAP [27]. Oligonucleotides are negatively charged and easy to encapsulate into cationic liposomes. Neutral liposomes are primarily constructed by neutral lipids, which include PC, PE, cholesterol, and DOPE [28]. Neutral liposomes have good biocompatibility and excellent pharmacokinetic characteristics, but they cannot interact with oligonucleotides to adsorb them and encapsulate them into the liposomes efficiently. Neutral liposomes are adopted to modified cationic liposomes to enhance particle stability. Ionizable liposomes are important for siRNA delivery. They can protonated and deprotonated according to the acidity of the environment [28]. Under hypoxic conditions, tumor tissues are more acidic and pH-responsive liposomes have more positive charges. Cationic liposomes are the most widely used form of liposomes.

Polymeric micelles have promising applications in drug delivery including extending the drug cycle time, changing the drug release curve, and easily connecting targeted ligands [29]. Cationic polymer micelles can ensure good oligonucleotide loading capacity through electrostatic adsorption. They show long circulation times, tumor passive targeting by the enhanced permeability and retention effect, and efficient oligonucleotide endosome release by the proton sponge effect [30]. Furthermore, the suitable carrier should can deliver oligonucleotides and chemotherapy drugs together to the tumor tissue and release the two drugs simultaneously, for example polymeric micelles with doxorubicin and siRNA targeting P-glycoprotein [31]. 

Nanoparticles using albumin, metals, and polymers have been used for drug delivery. Tumor cells can take up human serum albumin through endocytosis; therefore, albumin-based nanoparticles can show high stability without cytotoxicity [32]. Metallic nanometer-sized particles, such as silver, gold, and magnetic metals show the property of the enhanced surface to volume ratio; therefore, they have good applications in oligonucleotide delivery [33]. 

Another challenge to overcome in the DDS for pancreatic cancer is intratumoral injection [34] or implantation [35,36] of siRNAs in the pancreas (Figure 1). Implantation of Local Drug EluterR (LODER), can release siRNAs targeting KRAS over months in pancreatic cancer in vivo [36]. LODER is a biodegradable polymeric matrix that shields drugs against enzymatic degradation. EUS have enabled researchers to obtain pancreatic tissue samples and inject medicines into the pancreas repeatedly; therefore, DDS using EUS may improve the effectiveness of siRNA treatment for pancreatic cancer. In an animal model, we have reported that administration of siRNA by intratumoral injection with atelocollagen [37] and intravenous injection [38]. Both settings were effective to reduce targeted mRNA expression in vivo without severe side effects in the short term. Clinical trials are necessary to determine the long-term effects and safety of nucleic acid medicines.

## 6. Clinical Trials

### 6.1. Antisense Oligonucleotide

Eight nucleic acid medicines have been approved by the FDA (Table 4), five of which are ASOs used to treat nervous muscular diseases and familial metabolic diseases. 

There have been a lot of reports about ASOs for pancreatic cancer treatment in preclinical studies. *KRAS* is the most common target because approximately 90% of pancreatic cancer harbor *KRAS* mutation. AZD-4785, a high-affinity constrained ethyl-containing therapeutic ASO targeting *KRAS* mRNA, potently depleted *KRAS* mRNA in *KRAS*-mutant colon, pancreatic, and lung cancer cell lines, with no feedback activation of MAPK signaling. Significant antitumor activity was obtained in mice bearing *KRAS*-mutant lung cancer xenografts [39].

ASOs have been tested in more than 1000 clinical trials. Various ASOs have reached clinical trials for the treatment of pancreatic cancer. The targets of these molecules were related to cell proliferation (X-linked inhibitor of apoptosis protein, *XIAP* [40]; Protein Kinase A, *PKA* [41]), cell signaling (*HRAS*) [42], resistance to chemotherapy (heat shock protein 27, *Hsp27*) [43], or cancer stroma (*TGFβ2*) [44]. However, few ASOs have shown antitumor effects in clinical trials.

ISIS 2503 (ASO targeting *XIAP*) showed evidence of growth inhibition when combined with gemcitabine in locally advanced or metastatic pancreatic cancer in first-line treatment [40]. In that study, 58% of patients who received the combination survived 6 months or longer. Addition of apatorsen, the *Hsp27*-targeting antisense oligonucleotide, to chemotherapy did not improve outcomes in unselected patients with metastatic pancreatic cancer in the first-line setting, although a trend toward prolonged overall survival in patients with high baseline serum Hsp27 suggests that this therapy may warrant further evaluation in this subgroup.

### 6.2. Clinical Trials for siRNAs

Fourteen years after the first clinical trial using RNAi was entered (2004), the FDA approved the first therapeutic RNAi, ONPATTRO (patisiran), a lipid complex injection for treatment of peripheral nerve disease caused by hereditary transthyretin-mediated amyloidosis in adults [52] (Table 4). However, there is no clinically available therapeutic RNAi for pancreatic cancer.

Some siRNAs have already entered clinical trials for the treatment of locally advanced pancreatic cancer. siRNA targeting mutated *KRAS* is the most common [35,36]. The vast majority of *KRAS* mutations in pancreatic cancer are gain-of-function mutations, most of which occur in codon 12 with substitution of the Glycine for Aspartate (G12D). Golan et al. implanted siRNA targeting *KRAS* (G12D) in the pancreatic tumor using LODER in combination with Gemcitabine treatment [35]. The majority of patients (83%) demonstrated stable disease and 17% of patients showed partial response. Decrease in CA19-9 was observed in 70% of patients. The most frequent adverse events observed were grade 1 or 2 severity (89%); transient abdominal pain, diarrhea, and nausea. They concluded that the combination of mutated *KRAS*-targeting siRNAs and chemotherapy is well tolerated, safe, and demonstrated potential efficacy in pancreatic cancer patients [53]. 

Nishimura et al. have shown that EUS-guided fine-needle injection (EUS-FNI) of a synthetic double-stranded RNA oligonucleotide directed against *CHST15* (STNM01), an extracellular matrix component, was safe and feasible [34]. There were no adverse effects. STNM01 is also directly injected by endoscopy to treat ulcerative colitis. 

Atu027 is a liposomally formulated siRNA with antimetastatic activity, which silences protein kinase N3 (PKN3) expression in the vascular endothelium [54]. PKN3 acts as a Rho effector downstream of PI3K. Combination of Atu027 and gemcitabine for the treatment of advanced pancreatic cancer was safe and well tolerated. 

TKM-080301 is a lipid nanoparticle formulation of an siRNA against Polo-like kinase 1 (PLK1), which regulates critical aspects of tumor progression [55]. Preliminary antitumor efficacy for advanced pancreatic cancer has been observed. A potential molecular therapeutic context of increased PLK1 expression with inactivation of p53 or NF1 was observed in a remarkable responder. 

However, these data must be interpreted with caution because they are early-phase trials and some are still recruiting patients. The best responses observed so far have been tumor stabilization, with very few complete or partial responses documented. siRNAs were well tolerated but one death and a few grade 3–4 toxic effects due to elevation of liver enzymes were observed [56]. Several trials with different combinations including siRNAs are ongoing, and the combination of several nucleic acid medicines may be explored in the coming years. 

## 7. Conclusions

Despite the challenging obstacles, we hope that nucleic acid drugs will have a significant impact on the treatment of pancreatic cancer. The combination of genetic diagnosis using next-generation sequencing and targeted therapy may provide effective precision medicine for pancreatic cancer patients.

## Figures and Tables

**Figure 1 ijms-20-04224-f001:**
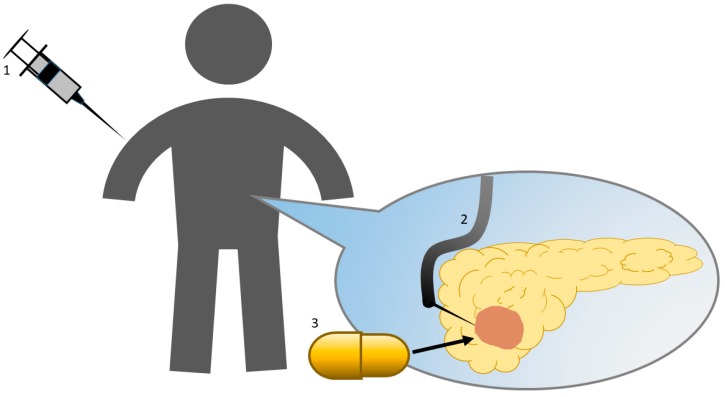
Delivery of nucleic acid medicines. (1) Intravenous injection, (2) intratumoral injection under EUS, and (3) intratumoral implantation.

**Table 1 ijms-20-04224-t001:** Nucleic acid medicines.

	Antisense Oligonucleotides	siRNAs	Antisense miRNAs	miRNA Mimics	Decoys	Aptamers	CpG Oligodeoxynucleotides
Structure	Single strand DNA/RNA	Double strand RNA	Single strand DNA/RNA	Double strand RNA	Double strand DNA	Single strand DNA/RNA	Single strand DNA
Length (base pairs)	12–21 20–30	20–25	12–16	20–25	20	26–45	20
Site	Intracellular (nucleus, cytoplasm)	Intracellular (cytoplasm)	Intracellular (cytoplasm)	Intracellular (cytoplasm)	Intracellular (nucleus)	Extracellular	Extracellular (endosome)
Target	mRNA pre-mRNA miRNA	mRNA	miRNA	mRNA	Protein (transcription factor)	Protein	Protein (TLR9)
Function	mRNA degradation Translational inhibition miRNA inhibition Splicing inhibition	mRNA degradation	miRNA degradation	mRNA degradation Translational inhibition	Transcriptional inhibition	Inhibition of protein function	Activation of natural immunity via TLR9
Drug delivery system	Modified or unnecessary	Necessary	Necessary	Necessary	Necessary	PEGylation	Antigen

TLR9, toll like receptor 9.

**Table 2 ijms-20-04224-t002:** Modifications of nucleic acid drugs.

Structural Modifications	Contents	Stability	Cellular Uptake	Gene Silencing Effect	Cytotoxicity	Binding Affinity
Diester modification	Phosphorothioate	superior	superior	inferior	superior	
Ribose modification	2’-O-Me, 2’-O-A, 2’-F	superior		inferior		
Base modification	Adenine methylation and deamination, cytosine methylation, hydroxy methylation and carboxy substitution, Guanine oxidation			superior		
Oligonucleotide analogues replacement	Peptide nucleic acid, locked nucleic acid, morpholino phosphamide	superior		superior		inferior
Conjugation to cell-penetrating peptides	Cysteine, transactivator of transcription peptide, gelatin		superior	superior	inferior	
Aptamer	20–100 nucleotides		superior	superior		

**Table 3 ijms-20-04224-t003:** Drug delivery systems.

	Materials
**Liposomes**	
Cationic liposome	DOTAP, DODMA, DOGS, DC-Chol
Neutral liposome	PC, Chol, DOPE
Ionizable liposome	DODMA, DODAP
**Micelles**	
Polymeric micelles	Amphiphilic copolymer, PEG, polyamino acid, polylactic or glycolic acid, polycaprolactone, and short phospholipid chains
Cationic polymer micelles	PEG-PLL-PLLeu, PEI-CG-PEI, PgP
**Nanoparticles**	
Albumin-based	thiol, arginine-glycine-aspartic acid peptide
Metal-based	gold, silver, magnetic

**Table 4 ijms-20-04224-t004:** Food and Drug Administration (FDA)-approved nucleic acid medicines.

Drug	Nucleic Acid	Disease	Modification	Administration	Company
Vitravene [45]	ASO	Cytomegalovirus retinitis	Phosphorothioated	Intravitreous	Isis Pharmaceuticals, Carlsbad, CA
Macugen [46]	Aptamer	Age-related macular degeneration	PEGylation 2’-F 2’-OMe	Intravitreous	Valeant Pharmaceuticals, Laval, Canada
Kynamro [47]	ASO	Homozygous familial hypercholesterolemia	Phosphorothioated 2’-MOE	Subcutaneous	Kastle Therapeutics, Chicago, IL
Exondys 51 [48]	ASO	Duchenne muscular dystrophy	Morpholino nucleic acid	Intravenous	Sarepta Therapeutics, Cambridge, MA
Spinraza [49]	ASO	Myelopathic muscular atrophy	Phosphorothioated 2’-MOE	Intraspinal	Biogen, Cambridge, MA
Heplisav-B [50]	CpG oligo	Hepatitis B	Phosphorothioated	Intramuscular	Dynavax Technologies, Berkeley, CA
Tegsedi [51]	ASO	Hereditary transthyretin-mediated amyloidosis	Phosphorothioated 2’-MOE	Subcutaneous	Akcea Therapeutics, Boston, MA
Onpattro [52]	siRNA	Hereditary transthyretin-mediated amyloidosis	2’-MOE	Intravenous	Alnylam Pharmaceuticals, Cambridge, MA

FDA, Food and Drug Administration; ASO, antisense oligonucleotide; CpG oligo, CpG oligodeoxynucleotide; 2’-MOE, 2’-O-methoxyethyl; 2’-OMe, 2’-O-Methyl; 2’-F, 2’-Fluoro.

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
