# Peer review of "Development and Clinical Trials of Nucleic Acid Medicines for Pancreatic Cancer Treatment"

_ijms, 2019, doi:10.3390/ijms20174224_

Round 1

Reviewer 1 Report

The manuscript entitled “Development and Clinical Trials of Nucleic Acid Medicines for Pancreatic Cancer Treatment” by Yamakawa, et al reviews current development of research and clinical trials on nuclear acid medicine. Overall, the manuscript is a comprehensive review. It is well written and covers many of the key topics in the selected area. The content in each section is detailed. However, there are several concerns, which need to be further addressed as follows.

1. The titles of each sections were not correctly numbered such as no number on introduction and 1 on all sections (1. Functions; 1. Modifications of nucleic acid drugs; 1. Aptamers; ….; 1. Conclusion).

2. Similarly, sub-sections were not numbered correctly such as 6-1. Antisense oligonucleotide and 6-1. Clinical trials for siRNAs.

3. Authors should add more information on siRNA and miRNA clinical trials. In ClinicalTrial.org, there are more than 70 registrations for siRNA and miRNA clinical trial. Some siRNAs such as Atu027 and TKM 080301 have been used for the clinical trials in pancreatic cancers. Author could discuss these developments.

4. Typos and grammar errors need to be corrected.

Author Response

Reviewer 1

The manuscript entitled “Development and Clinical Trials of Nucleic Acid Medicines for Pancreatic Cancer Treatment” by Yamakawa, et al reviews current development of research and clinical trials on nuclear acid medicine. Overall, the manuscript is a comprehensive review. It is well written and covers many of the key topics in the selected area. The content in each section is detailed. However, there are several concerns, which need to be further addressed as follows.

The titles of each sections were not correctly numbered such as no number on introduction and 1 on all sections (1. Functions; 1. Modifications of nucleic acid drugs; 1. Aptamers; ….; 1. Conclusion).

Response: Thank you for your comments. Previous version of our manuscript had correct numbering for titles while submitting; however, the system of the journal might have changed them. We have ensured correct numbering in the revised manuscript as well.

Similarly, sub-sections were not numbered correctly such as 6-1. Antisense oligonucleotide and 6-1. Clinical trials for siRNAs.

Response: Previous version of our manuscript had correct numbering for sub-sections; however, the system of the journal might have changed them.

Authors should add more information on siRNA and miRNA clinical trials. In ClinicalTrial.org, there are more than 70 registrations for siRNA and miRNA clinical trial. Some siRNAs such as Atu027 and TKM 080301 have been used for the clinical trials in pancreatic cancers. Author could discuss these developments.

Response: We have added the information on the clinical trials of siRNAs for pancreatic cancer and discussed it.

Typos and grammar errors need to be corrected.

Response: We have corrected typos and grammar errors.

Reviewer 2 Report

The authors have discussed a revolutionary, yet relevant topic that is not only applicable to researchers of pancreatic cancer, but to cancer researchers in general. The authors review the existing research on nucleic acid-based drugs and drug delivery systems to target specific mutations in cancer for more efficacious therapy. They exhaustively cover the various NA-based drugs, briefly discuss their functions, present the challenges for each of the drug classes and put forth their ideas for furthering research to improve the drug performances. 

I would urge the authors to consider the following points to revise their manuscript to improve readability and organization:

1. Abstract, Line 29: "hurbor drugable" to "harbor targetable"
2. What do they mean by "drugable" mutations? Do these have
clinically approved drugs in other cancers?
3. Line 41, provide reference or at least discuss why side-effects "might" be lower.
4. Please provide references in section for ASOs (Lines 58-67). Furthermore, the authors should also bring the functional mechanisms of aptamers, decoys and CpG oligos as done for ASOs and siRNA.
5. Subsections "2-1. Antisense oligoneucleotide" and "2-1. siRNA" both are misnumbered.

6. Sections following Introduction all have the same number "1"
7. Table 2, the authors don't specify what the "up"s and "down"s are in comparison to?
8. Line 198, the authors probably meant "challenge" instead of "change"
9. Table 4 should have an additional column citing the clinical trials

Author Response

Reviewer 2

The authors have discussed a revolutionary, yet relevant topic that is not only applicable to researchers of pancreatic cancer, but to cancer researchers in general. The authors review the existing research on nucleic acid-based drugs and drug delivery systems to target specific mutations in cancer for more efficacious therapy. They exhaustively cover the various NA-based drugs, briefly discuss their functions, present the challenges for each of the drug classes and put forth their ideas for furthering research to improve the drug performances.

 I would urge the authors to consider the following points to revise their manuscript to improve readability and organization:

Abstract, Line 29: "hurbor drugable" to "harbor targetable"

Response: Thank you for your comments. We have corrected it.

What do they mean by "drugable" mutations? Do these have clinically approved drugs in other cancers?

Response: We meant that they have clinically or experimentally approved drugs in other cancers.

Line 41, provide reference or at least discuss why side-effects "might" be lower.

Response: We have added reference.

Please provide references in section for ASOs (Lines 58-67). Furthermore, the authors should also bring the functional mechanisms of aptamers, decoys and CpG oligos as done for ASOs and siRNA.

Response: We have added reference and the functional mechanisms of aptamers, decoys and CpG oligos.

Subsections "2-1. Antisense oligoneucleotide" and "2-1. siRNA" both are misnumbered.

Response: Previous our manuscript had correct number of sub-sections, but the system of the journal might change them.

Sections following Introduction all have the same number "1"

Response: Previous our manuscript had correct number of the titles, but the system of the journal might change them. We have ensured correct numbering in the revised manuscript as well.

Table 2, the authors don't specify what the "up"s and "down"s are in comparison to?

Response: We have changed "up"s and "down"s to "superior"s and "inferior"s.

Line 198, the authors probably meant "challenge" instead of "change"

Response: We have corrected it.

Table 4 should have an additional column citing the clinical trials

Response: We have added the citations

Round 2

Reviewer 1 Report

the authors have addressed most of the critiques

Reviewer 2 Report

Thank you for addressing my concerns